# Cell-size dependent progression of the cell cycle creates homeostasis and flexibility of plant cell size

Angharad R. Jones[1], Manuel Forero-Vargas[1,†], Simon P. Withers[1], Richard S. Smith[2], Jan Traas[3], Walter Dewitte[1,\*] & James A.H. Murray[1,\*]

Mean cell size at division is generally constant for specific conditions and cell types, but the mechanisms coupling cell growth and cell cycle control with cell size regulation are poorly understood in intact tissues. Here we show that the continuously dividing fields of cells within the shoot apical meristem of Arabidopsis show dynamic regulation of mean cell size dependent on developmental stage, genotype and environmental signals. We show cell size at division and cell cycle length is effectively predicted using a two-stage cell cycle model linking cell growth and two sequential cyclin dependent kinase (CDK) activities, and experimental results concur in showing that progression through both G1/S and G2/M is size dependent. This work shows that cell-autonomous co-ordination of cell growth and cell division previously observed in unicellular organisms also exists in intact plant tissues, and that cell size may be an emergent rather than directly determined property of cells.

[1] Division of Molecular Biosciences, Cardiff School of Biosciences, Cardiff University, Cardiff, Wales CF10 3AX, UK. [2] Department of Comparative Development and Genetics, Max Planck Institute for Plant Breeding Research, 50829 Cologne, Germany. [3] Laboratoire de Reproduction de développement des plantes, INRA, CNRS, ENS Lyon, UCB Lyon 1, Université de Lyon, 69364 Lyon Cedex 07, France. † Present address: Grupo de Investigatión D + TEC, Facultad de Ingeniería, Universidad de Ibagué, 730002 Ibagué, Colombia. \* These authors jointly supervised this work. Correspondence and requests for materials should be addressed to J.A.H.M. (email: MurrayJA1@Cardiff.ac.uk).

Cell size depends on the two opposing processes of growth and division. To maintain a constant distribution of cell sizes over generations, cells must be neither too large nor too small when they divide. If growth is linear, this can be achieved simply by dividing symmetrically after a constant amount of time, but if growth is exponential or cells do not divide symmetrically, cell size must be actively maintained and division triggered by size rather than age[1,2]. According to such 'sizer' models, large cells will divide faster than small cells, a prediction that has been confirmed in yeasts by comparing populations of cells of different sizes produced by varying growth conditions[3,4], by inducing temporary blocks to cell cycle progression[5] or by utilizing naturally occurring asymmetric divisions[4,6].

Size control is generally considered to occur at one or more of the two primary cell cycle control checkpoints that precede the initiation of DNA synthesis (G1/S transition) and the onset of nuclear division (G2/M transition), and a single cycle may consist of a combination of sizer and timer steps[7–9]. Furthermore, the critical size required for cell cycle progression is dependent on environmental conditions[3,10–12], therefore any underlying mechanism must not only explain size homeostasis, but also allow for environmental adaptation of cell size[12–14].

Although many theoretical models have been proposed, identifying the molecular mechanisms behind cell size control has been more difficult. The critical cell size required for division may be directly measured using a 'molecular ruler' such as Pom1 (refs 15,16), an inhibitor of cell division localized to the ends of rod-shaped fission yeast cells that blocks entry to mitosis until cells have reached a critical length. Alternatively, mean cell size at division may be an emergent property of a system in which the accumulation[2,7,17,18], dilution[2,19] or destruction[20] of a protein, usually involved in the regulation of a particular phase transition of the cell cycle, is proportional to cell size. In budding yeast, size-dependent production of the positive G1/S regulator cyclin Cln3 has been proposed as such a size-control mechanism[21], but more recently dilution of the negative cell cycle regulator Whi5 through cell growth has been suggested as a more likely mechanism[19]. In both fission yeast[12,13] and budding yeast[14], the critical size for division is set according to nutrient availability via the conserved TOR signaling pathway which feeds into the activity of key cell cycle regulators.

It is less clear whether such intrinsic cell size control is likely to play a large role in regulating cell size in multicellular organisms[22,23], where cell size may be constrained by tissue structure and changes in cell size are associated with development and morphogenesis. Indeed extracellular signals that play roles in co-ordinating development have been shown to be essential for growth and division of higher eukaryotic cells[22–24], indicating that cell size may be primarily regulated by mechanisms that operate at the level of the tissue. Answering this question experimentally has been particularly difficult since significant technical challenges are associated with transferring techniques from yeast to higher eukaryotes, particularly if positional and developmental information is to be retained. Studies using mammalian cell cultures have produced conflicting results[25–30], but recent technical advances suggest that cell growth is not linear[28–30] and therefore active control of cell size is required, although the mechanism is not yet clear.

In plants, cell division is largely restricted to meristematic regions of the root and shoot. The shoot apical meristem (SAM) is a complex domed structure that houses the stem cell niche and initiates above-ground organs (leaves and flowers) on its flanks. The structure is accessible through dissection and continues to undergo development for several days in culture[31,32]. Studies to date show that cell size in the SAM is tightly developmentally regulated, with smaller cells in the central zone, where the stem cell niche is located and larger cells in developing organs[33].

SAM cells are subject to tissue level controls from the plant hormones auxin and cytokinin, which are essential for cell growth and division[24], as well as to mechanical constraints that arise from cells being connected via semi-rigid walls and affect the plane of cell division[34]. Analysis of cell growth rates in the SAM suggests cell size control is required[35] and indeed most models of plant tissue growth assume that cell division is triggered when cells reach a defined cell size[36–39]. However no cell-sizing mechanism has been proposed or identified for plant cells, and little is known about how cells behave when variation in cell size arises, or how changes in cell size are regulated during development. Furthermore, it is not known when or how cell size is integrated with the control of the plant cell cycle.

Here we use four-dimensional (4D) time course analysis of developing SAMs to address the question of cell size control in plants, both in the meristematic central zone and the developing primordia. We investigate the relationship between cell size and cell cycle progression in this intact, growing tissue and identify at which points in the cell cycle size information is likely to be processed. Using a predictive model linking growth rate and cell cycle regulation, we probe the effects of altering cell cycle regulators on cell size and test these predictions using mutant lines. Our results reveal dynamic regulation of cell size at both major cell cycle checkpoints in plant cells, and are consistent with cell size at division being an observed emergent property.

## Results

**Cell size is controlled by varying cell cycle length.** We first investigated the growth of cells in living intact meristems. Four Arabidopsis meristems were imaged by mounting dissected inflorescences in a confocal scanning microscope and scanning every 8 h over a total of 96 h. Curved surface projections (2.5D) of the outer cell layer (L1) were used to segment the images and identify cells and lineages[40]. Since the L1 is a uniform thickness, the outer cell surface area is a good proxy for cell volume (Supplementary Fig. 1a–c). In agreement with previous work[35], we found that growth of both individual cells and groups of cells is non-linear, with the absolute increase in cell area increasing over time as the cells grow (Fig. 1a,c). Indeed, taking the natural logarithm of the increase in area indicates that growth approximates to exponential growth (Fig. 1b,d). Since it has been demonstrated that non-linearly growing cells must have cell size control to prevent differences in size from being amplified over generations[1], we agree with previous studies[35] in concluding that a cell size control mechanism must be operating in the SAM. Again in agreement with previous analyses[35], we found only a weak relationship between cell size and the relative growth rate (RGR) of the cell (Supplementary Fig. 1d), indicating that cell size control is not achieved simply by restricting the growth of the largest cells. In contrast, we found a strong inverse relationship between cell size at birth and cell cycle length (Fig. 1e,f) such that smaller cells take longer to divide than larger cells. These results suggest a linkage between cell cycle length and cell size which may account for cell size control[1].

To further investigate whether the relationship between cell size and cell cycle length is responsible for cell size control, we used the occurrence of unequal divisions to observe how variation in cell size is removed. Consistent with previous studies[38,35], we found that ~60% of divisions produced daughters with >5 μm² difference in outer cell surface area at birth, equivalent to inheriting on average >54% of the parental area (Supplementary Fig. 2a,b). In the central zone, 182 divisions were identified where both daughters could be tracked through an entire division cycle

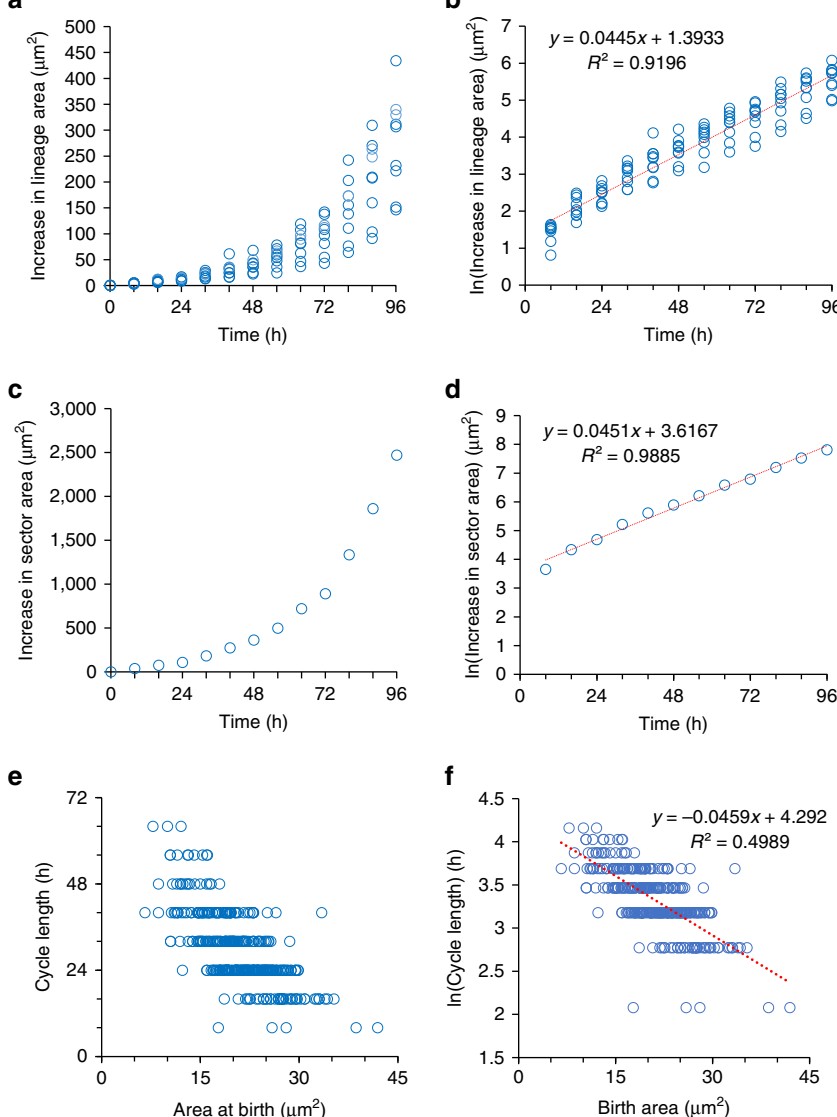

**Figure 1 | Cell size control is required to maintain cell size in the SAM.** (**a**) Graph showing the increase in lineage area of the nine most central cell lineages in the central region of the SAM over 96 h of growth. Each lineage begins with a single cell. (**b**) Natural log transformed data from graph **a**. Linear regression indicates an exponential relationship. (**c**) Increase in total area of the lineages shown in parts **a**,**b**. (**d**) Natural log transformed data from graph **c**. Linear regression indicates an exponential relationship. (**e**) Graph showing relationship between cell area at birth and cell cycle length for 182 cells from the central zone. (**f**) Natural log transformed data from graph **e**. Red line shows linear regression.

(Fig. 2a–c). In 46 cases (23%), daughters divided synchronously, but in most cases (125/182) daughters divided asynchronously with up to 48 h difference in cell cycle length (Fig. 2b–d). Notably, sisters that divided asynchronously had a larger mean difference in area at birth than those that divided synchronously (Fig. 2e) and in all but eight cases (4% of asynchronous divisions), the larger daughter divided first and on average added a smaller area to its birth size (Fig. 2e). Accordingly, a significant number of asymmetric pairs ($>5\,\mu m^2$ difference in area) were closer in size at division than at birth (binomial test, 72/105, $P = 3.885e^{-05}$). Differences in size at birth therefore appear to be corrected mainly by changes to cell cycle length through an inverse relationship that results in larger, faster cycling cells and smaller, slower cycling cells.

**Cell size depends on developmental stage and environment.** Using a further three stems imaged every three hours over a 30-h

time course, we detected the same inverse relationship between birth size and cell cycle length throughout primordium development, during which mean cell size increases (Fig. 2f). However, whereas large and small daughters produced by uneven divisions showed small differences in RGR (Supplementary Fig. 2c), mean RGR increased significantly during primordium formation (Fig. 2g). Cells in developing primordia consequently added more material per cycle than slower growing cells in the central region and maintained larger cell sizes despite their shorter cell cycle. Our results therefore indicate that although homeostasis of cell size is produced by the inverse relationship between cell size and cell cycle length, the homeostatic cell size is itself dependent on RGR.

In unicellular organisms, similar relationships between RGR and cell size are thought to be the result of metabolic constraints[3,10–12]. We therefore investigated the effect of environmental conditions that restrict photosynthesis on cell size. Plants were grown to the floral transition under normal light

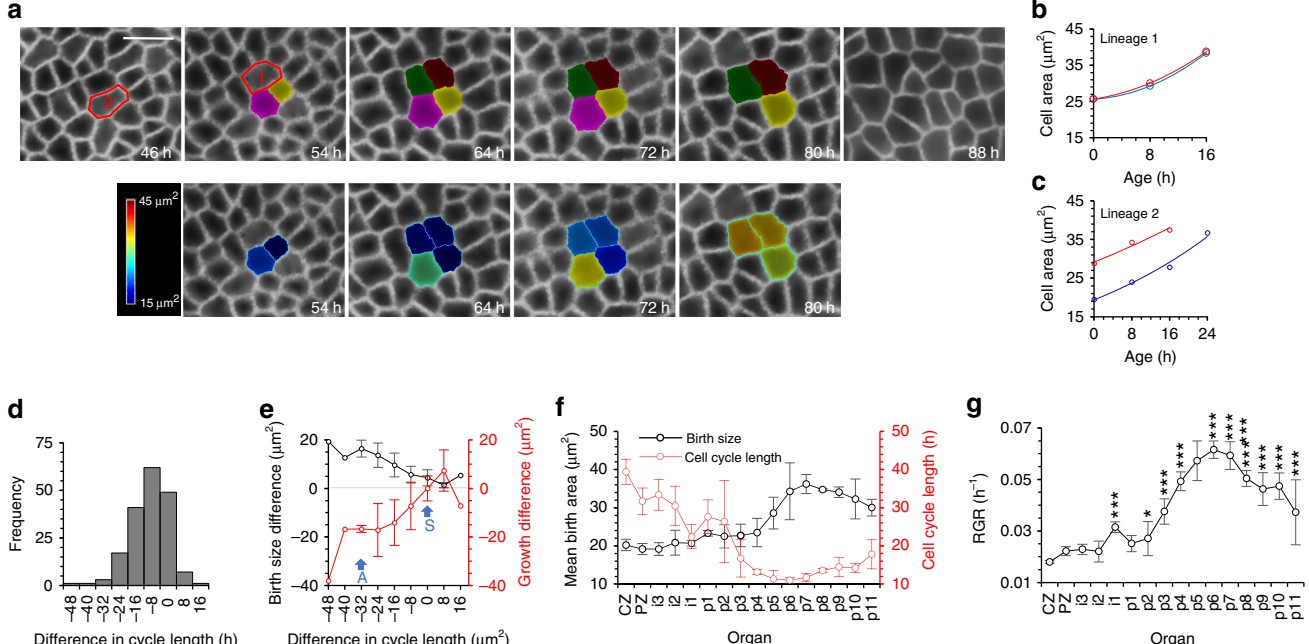

**Figure 2 | An inverse relationship between cell size and cell cycle length and a positive relationship between RGR and cell size regulate cell size in the SAM and developing primordia** (**a**) Time course images showing two example lineages (labelled 1 and 2) where both daughters of a division can be tracked through an entire cycle. Colours indicate cell identities (upper row) and outer cell surface area (lower row). Lineage 1 shows equal division of the parent cell. Lineage 2 shows unequal division of the parent cell. Scale bar represents 10 μm. (**b,c**) Cell sizes over time of daughters of lineages 1 and 2, which divide synchronously and asynchronously respectively. The daughter with the largest birth size is shown in red and the daughter with the smaller birth size is shown in blue. (**d**) Frequency distribution of differences in cell cycle length between daughter cells from the same division, calculated as the cycle length of the largest daughter at birth minus the cycle length of the smallest daughter at birth. $n = 182$. (**e**) Mean difference in size at birth (black), calculated as area of larger daughter at birth—area smaller daughter at birth, and mean difference in total growth (red) calculated as increase in area of larger daughter at birth minus the increase in area of smaller daughter at birth. Negative growth difference values indicate that the smaller daughter grew by the largest amount. Data grouped by difference in cell cycle length ($n$ varies by group, $n = 1, 1, 3, 17, 41, 62, 49, 7, 1$). Synchronously (S) dividing sisters show little difference in size at birth or in growth, but asynchronously dividing sister cells (A) show larger differences in size at birth and undergo different amounts of growth. Error bars represent s.d. (**f**) Mean cell size at birth and mean cell cycle length of cells grouped by developmental zones where CZ = central zone, PZ = peripheral zone, i = incipient primordia and P = primordia. Means calculated from three stems. Error bars represent s.d. (**g**) Mean relative growth rate (RGR) of cells grouped by developmental zones as described above. Means calculated from three stems. Error bars represent s.d. Note that there is greater variation between cells in different zones than between cells in the same zones (One-way ANOVA, $F(15,1) = 29.989$, $P < 0.001$). *, **, *** indicate significant difference from the RGR of the CZ at the 0.05, 0.01 and 0.001 levels respectively.

intensity, then transferred to low-light conditions. We observed a decrease in the overall size of the SAM and number of primordia produced, but, in addition to these higher-level adaptations, a small but significant reduction in outer cell surface area was observed (effect size = 2.81 μm² ± 0.7, $P < 0.001$) (Fig. 3a–c). Cell size could be restored by returning plants to normal light conditions (effect size = 2.54 μm² ± 0.88, $P < 0.001$) (Fig. 3d–f) or culturing low-light grown apices on media containing 1% sucrose (effect size = 8 μm² ± 0.67, $P < 0.001$)(Fig. 3g–j). These results indicate that the observed change in cell size is a dynamic response to carbon source availability. Division size is therefore not fixed at an absolute value, but instead is dynamic and dependent on both developmental status and, as in unicellular organisms, on metabolic constraints.

**One transition model of the cell cycle.** The dynamic nature of regulation suggests that maintenance of cell size may be the result of the interplay of cell growth rate and cell cycle length rather than being triggered at a fixed, absolute size. In this case, to produce larger faster cycling cells and smaller slower cycling cells, cell cycle progression itself must be dependent on cell size. As in all other eukaryotes, the plant cell cycle consists of two active phases, synthesis (S) and mitosis (M), separated by two gap

phases (G1 and G2). Progression is regulated by the accumulation of CYCLIN DEPENDENT KINASE (CDK) activity to threshold levels that trigger the G1/S and G2/M transitions. Regulation of the production of CDK activity[21] or the CDK threshold for transition[19] have both been proposed as plausible mechanisms for integration of cell size into cell cycle progression in yeast[2]. To explore whether such a mechanism might explain our experimental observations, we produced a model of CDK accumulation in a growing cell that considers the relative growth rate of the cell ($g$), the rate of production of active CDK (pCDK) and the threshold level of CDK ($T_{Division}$) required for division (Fig. 4a). In yeast it has been demonstrated that the G1/S and G2/M transitions are driven by the overall CDK level, with a higher threshold required for the G2/M transition[41,42]. For simplicity we therefore assumed that activity of a single CDK with constant pCDK is sufficient to drive cell cycle progression, and, at a threshold level ($T_{Division}$) triggers cell division according to a division ratio $d$. Thus in contrast to existing models of plant tissue growth, division size is determined by the size at which the CDK threshold is met, rather than being specified as a fixed geometric parameter.

To explain the experimental data, the model must: (i) maintain a constant average division size over generations, (ii) remove heterogeneity created by uneven divisions, (iii) predict an inverse

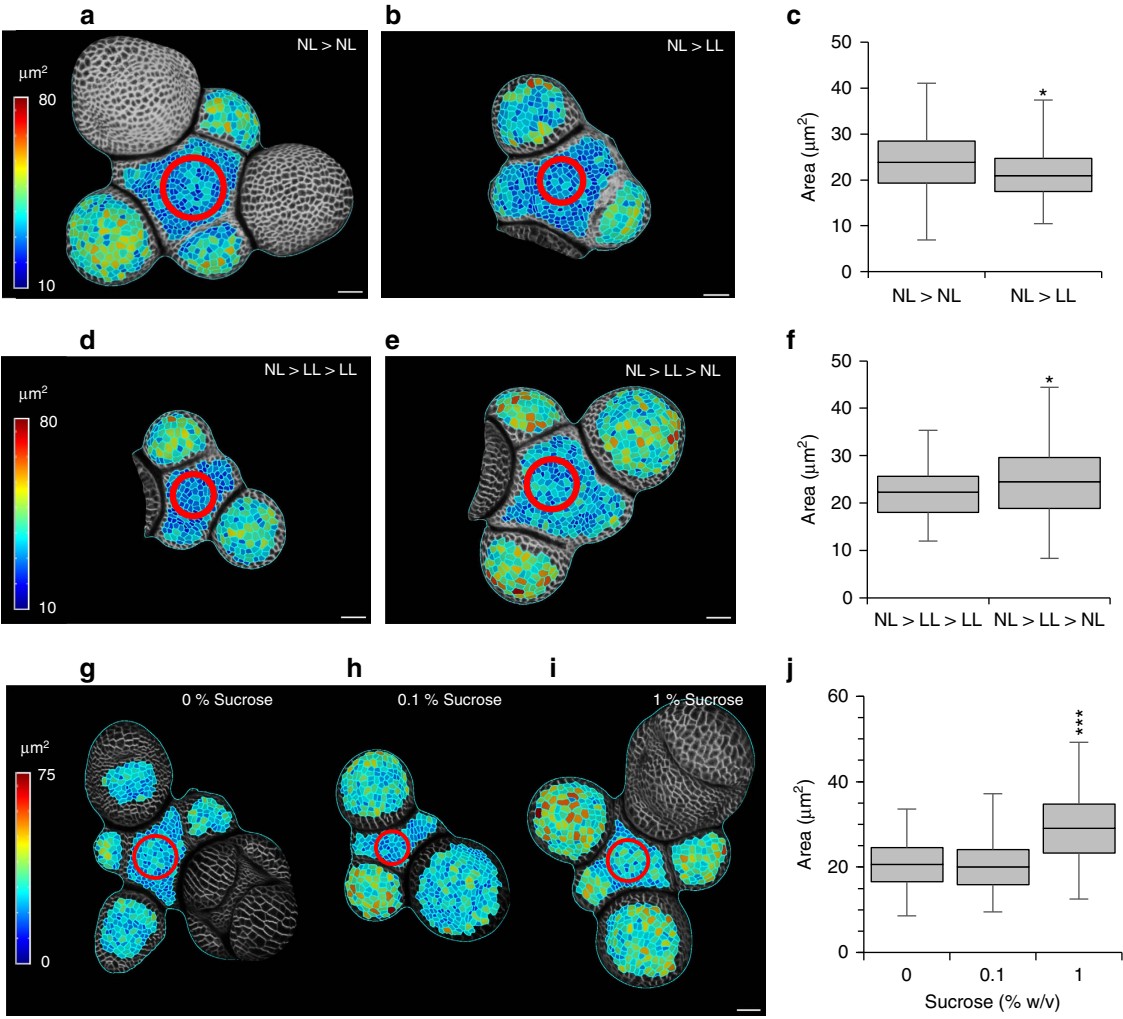

**Figure 3 | Mean cell size in the SAM is dynamic and dependent on metabolic constraints.** (**a,b**) Segmented surface projections of stems from plants grown to floral transition under normal light intensity (NL) then either kept under NL conditions (**a**) or transferred to low light intensity (LL) conditions (**b**). Shading indicates outer cell surface area. Red circle indicates the region between outgrowing primordia used for quantification. Scale bars represent 20 μm. (**c**) Distribution of cell sizes under NL > NL and NL > LL. Boxes represent the interquartile range, whiskers represent total range. (Generalized Linear Mixed Model (GLMM), $n = 348$ cells from 12 stems, 6 stems per treatment). (**d,e**) Segmented surface projections of stems grown under NL > LL conditions then kept in LL (**d**) or transferred back to NL (**e**). Shading indicates the outer cell surface area. Red circle indicates the region between outgrowing primordia used for quantification. Scale bars represent 20 μm. (**f**) Distribution of cell sizes under NL > LL > LL and NL > LL > NL conditions. Boxes represent the interquartile range, whiskers represent total range. (GLMM, $n = 321$ cells from 10 stems, 5 stems per treatment). (**g–i**) Segmented surface projections of dissected stems from plants grown under NL > LL conditions, then cultured for 72 h growth on media containing 0% (**g**), 0.1% (**h**) or 1% sucrose (**i**). Shading indicates the outer cell surface area. Red circle indicates the region between outgrowing primordia used for quantification. (**j**) Distribution of cell sizes following 72 h growth on medium containing 0, 0.1 or 1% sucrose. Boxes represent the interquartile range, whiskers represent total range. (GLMM, $n = 483$ cells from 12 stems, 3 stems per treatment). *, **, *** indicate a significance at the 0.05, 0.01 and 0.001 levels respectively.

relationship between birth size and cycle length and (iv) predict division size proportional to $g$. We first ran the model without variation in $g$ or $d$ and without any interaction between cell size and pCDK or $T_{\text{Division}}$. In this case, if neither pCDK nor $T_{\text{Division}}$ are dependent on cell size, cell-cycle length is invariant, consistent with a 'timer' mechanism[1]. In this case cell size at division is only constant when cell cycle length is equal to the time taken for the cell to double in size. Cell size does not converge following a simulated uneven division (Fig. 4b) and changing the RGR ($g$) leads to unstable division sizes (Fig. 4c). If experimentally observed variations in RGR and division ratio within a population are included in $g$ and $d$, the distribution of cell sizes degrades over time, leading to an increasingly broad distribution of cell sizes (Fig. 4d). The model demonstrates that a timer mechanism based on the parameters measured from our experiments would not be sufficient to produce constant cell size.

In contrast, if pCDK is proportional to cell size (Fig. 4e), such that the bigger the cell the more active CDK it produces, division size is more stable. This represents an initiator-accumulator 'sizer' mechanism where accumulation of CDK activity acts as the proxy for cell size[2,18]. As expected of a 'sizer' model, division size returns to normal within a few cycles of an uneven division because the large cell divides more rapidly than the small cell (Fig. 4f). Also, consistent with our experimental data, cell size is dependent on the RGR $g$, with a higher rate resulting in a larger cell size at division and a faster cell cycle (Fig. 4g). The model furthermore remains stable when noise in division sizes is introduced, predicting a steady distribution of cell sizes within the population over time (Fig. 4h) within which the inverse relationship between cell size at birth and cell cycle length is observable at the population level (Fig. 4i). Cell size control was produced whether pCDK was dependent on Size, Size$^{0.67}$ or

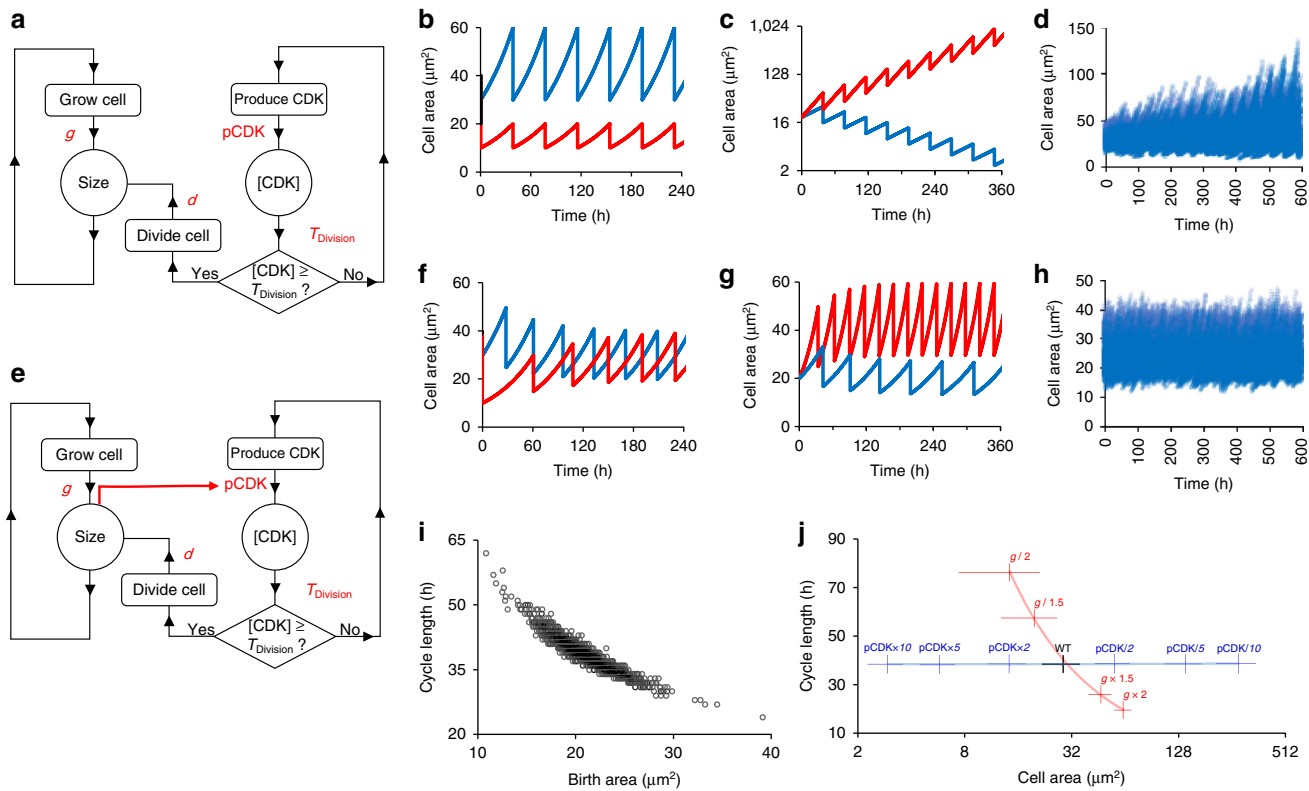

**Figure 4 | Cell size dependent progression of the cell cycle is sufficient to produce an inverse relationship between cell size and cell cycle length and homeostasis of cell size.** (**a**) Schematic of model wiring. Model consists of a growth loop (left hand side) and a CDK loop (right hand side). With each time step, size and CDK concentration are recalculated according to the relative growth rate (*g*) and the rate of production of active CDK (pCDK), respectively. When the concentration of CDK rises above the threshold level ($T_{Division}$) cell division is triggered and cells divide according to a division ratio *d*. (**b,c**) Simulation results for model where pCDK and $T_{Division}$ are independent of cell size. Simulations are run without variation in *g* or *d* and are designed to test the effect an uneven division (**b**) or a change in RGR (**c**). In the uneven division test the simulation is initiated with a large cell (blue) and a small cell (red). In the RGR test, the simulation was run with increased (red) or decreased (blue) relative growth rate *g*. Graphs show cell size over time, representing a period of at least six division cycles. (**d**) Simulation where pCDK and $T_{Division}$ are independent of cell size, including variation in *g* and *d* according to observed values. The population is initiated with 100 asynchronous cells. (**e**) Schematic of model wiring. Growth loop and CDK loop are integrated by making the production of CDK dependent on size (red arrow). (**f,g**) Simulation results where pCDK is directly proportional to cell size. Simulations of an uneven division (**f**) and a change in RGR (**g**) were run as above (**b,c**). (**h**) Simulation using model with pCDK directly proportional to cell size, including variation in *g* and *d* according to observed values. The population is initiated with 100 asynchronous cells. (**i**) Relationship between cell size and cell cycle length in individual cells in a simulated population of 100 asynchronous cells. (**j**) The effect of altering RGR (red) and pCDK (blue) on the distribution of cell sizes and cell cycle lengths in a population of 100 cells simulated as above. $T_{Division}$ is unaltered. Relative fold changes in parameter values relative to WT values are indicated on the graphs. Note that altering pCDK changes cell size, but not cell cycle length. Data points represent means, error bars represent s.d.

Size$^{0.34}$ indicating that pCDK and Size need not have the same dimensions in order to produce homeostasis. However, control is less efficient as a greater number of cycles are required to return the system to a stable state after perturbation if Size$^{0.67}$ or Size$^{0.34}$ relationships are used. (Supplementary Fig. 3a–f). Similar results are obtained if $T_{Division}$ is inversely proportional to cell size according to an inhibition-dilution sizer model[2] (Supplementary Fig. 4a–c), but not if pCDK is inversely proportional to cell size or $T_{Division}$ is proportional to cell size (Supplementary Fig. 4d–i) in which cases the relationship between cell size and cell cycle length is reversed. Size-dependent progression of the CDK cycle, producing larger faster cycling cells and smaller slower cycling cells, is therefore sufficient to predict our experimental observations.

**The CDK cycle co-ordinates cell size and cell cycle length**. To better understand the role of the CDK cycle in mediating the relationship between cell size and cell cycle length, we re-ran the model to predict the distribution of cell sizes in the meristems of mutants with a higher or lower rate of CDK production (pCDK). pCDK had a strong effect on the distribution of cell sizes; decreasing pCDK increased predicted cell sizes, whereas increasing pCDK led to decreased cell sizes (Fig. 4j). Despite the predicted changes in cell size, cell cycle length did not change regardless of whether pCDK or $T_{Division}$ was used as the size-dependent parameter. This is in notable contrast to the effect of altering *g*, which affects both cell size and cell cycle length (Fig. 4j). This reflects that the CDK cycle acts as the 'gearing mechanism' between cell size and cell cycle length. pCDK must therefore be constant in order to maintain the conversion of increased cell size into a more rapid cell cycle.

Although there are a variety of mechanisms known to control the level of CDK activity, protein synthesis is central: at G1/S due to the lability of the regulatory subunit CYCLIND (CYCD)[43], which is rate limiting for the G1/S transition[44] and has been shown to bind to CDKA (ref. 44) and increase CDKA activity[45,46], and at G2/M due to cell cycle dependent

expression of both CYCLINB (refs 47,48) and CDKBs[49–51], a plant-specific group of mitotic CDKs with rate-limiting activity at the G2/M transition[52].

To test our model predictions regarding pCDK, we measured cell sizes in meristems of mutants in which synthesis of CYCD3 and CDKB1 proteins are altered[53–55], and also determined mean cell cycle length. Mean cell size was increased both in the *cycd3;1-3* mutant lacking all three *CYCD3* genes[54] (Fig. 5a–c) and the *cdkb1;1/1;2* mutant[55] (Fig. 5i–k) lacking both *CDKB1* genes, which are likely to reduce G1/S and G2/M CDK activity respectively, equivalent to reducing pCDK in our model. Conversely, mean cell size was reduced when *CYCD3;1* was overexpressed throughout the cell cycle under the control of the CaMV35s promoter (Fig. 5e–g), equivalent to increasing pCDK in our model. Despite changing cell size significantly, neither the mutants nor overexpressor produced a significant change in RGR (Supplementary Fig. 5a-c) or cell cycle length (Fig. 5d,h,l) compared to wild type. This is consistent with the predicted effect of altering the value of pCDK in the model and supports the proposal that CDK activity is essential in mediating the relationship between cell size and cell cycle length.

**Two transition model of the cell cycle.** In a study of fixed cells from Arabidopsis floral primordia, most growth was reported to take place during G1 (ref. 56), leading to the suggestion that, as in

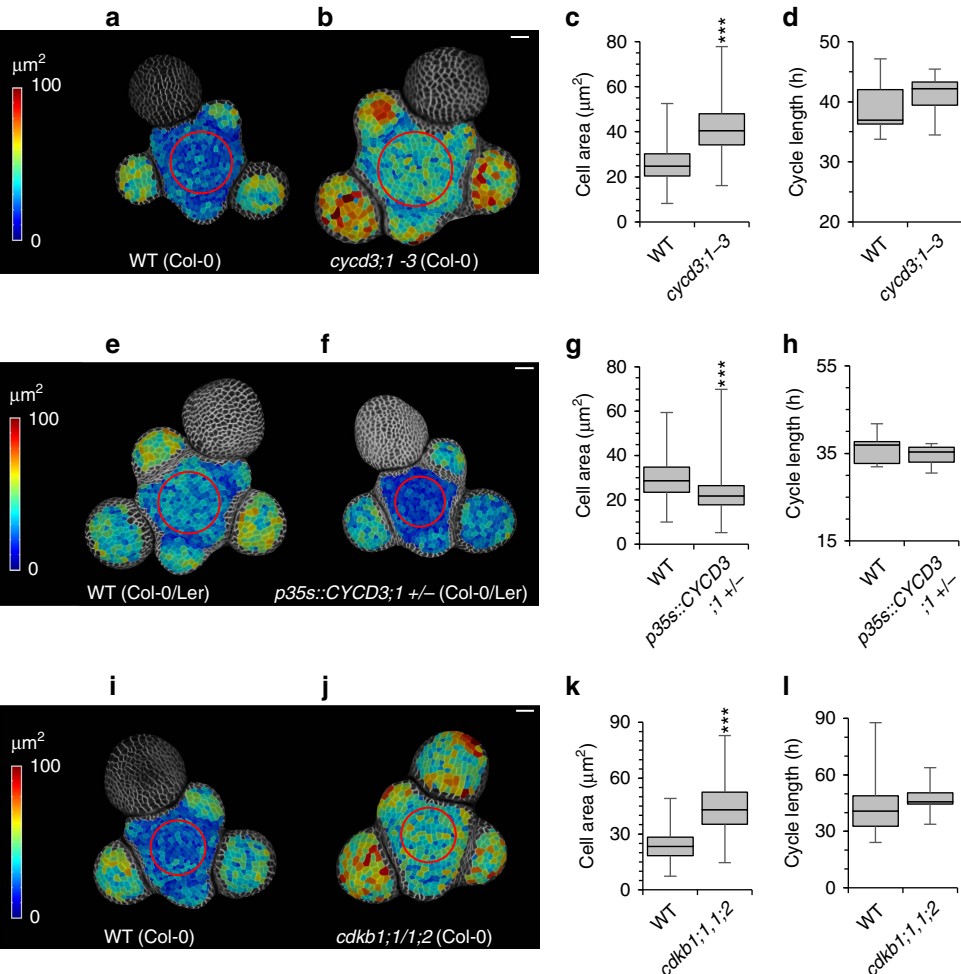

**Figure 5 | Production of active CDK affects cell size but not cell cycle length.** (**a,b**) Segmented surface projections of stems of wild type (Col-0) (**a**) and *cycd3;1-3* triple mutant plants (**b**). Shading indicates outer surface area. Scale bars represent 20 μm. (**c**) Distribution of cell sizes in WT (Col-0) and *cycd3;1-3* triple mutant plants. Boxes represent the interquartile range, whiskers represent total range. (GLMM, WT = 1820 cells from 9 stems, *cycd3;1-3* 1482 cells from 8 stems, Effect size = 15.87 μm² ± 0.61, P<0.001) (**d**) Distribution of mean cell cycle length in WT (Col-0) and *cycd3;1-3* triple mutant plants. Boxes represent the interquartile range, whiskers represent total range. (*t*-test, WT 9 stems, *cycd3;1-3* 8 stems, t = 0.867, df = 13.5, p 0.4011) (**e,f**) Segmented surface projections of stems of wild type (Col-0/Ler) (e) and *35s::CYCD3;1 +/−* plants (**f**). Shading indicates outer surface area. Scale bars represent 20 μm. (**g**) Distribution of cell sizes in WT (Col-0/Ler) and *35s::CYCD3;1±* plants. Boxes represent the interquartile range, whiskers represent total range. (GLMM, WT 769 cells from 5 stems, *35s::CYCD3;1±* 1320 cells from 7 stems, Effect size 4.73 μm² ± 0.39, P<0.001) (**h**) Distribution of mean cell cycle length in WT (Col-0/Ler) and *35s::CYCD3;1±* plants. Boxes represent the interquartile range, whiskers represent total range. (*t*-test, WT 5 stems, *35s::CYCD3;1-3* 7 stems, t = − 0.7899, df = 6718, P = 0.4566) (**i,j**) Segmented surface projections of stems of WT (Col-0) (**i**) and *cdkb1;1/1;2* double mutant plants (**j**). Shading indicates outer surface area. Scale bars represent 20 μm. (**k**) Distribution of cell sizes in WT (Col-0) and *cdkb1;1/1;2* double mutant plants. Boxes represent the interquartile range, whiskers represent total range. (GLMM, WT = 894 cells from 13 stems, *cdkb1;1/1;2* = 593 cells from 11 stems, Effect size = 20.24 μm² ± 0.53, P<0.001) (**l**) Distribution of mean cell cycle length in WT (Col-0) and *cdkb1;1/1;2* double mutant plants. Boxes represent the interquartile range, whiskers represent total range. (*t*-test, WT = 13 stems, *cdkb1;1/1;2* = 11 stems, t = 0.6448, df = 16.766, P = 0.5278) *, **, *** indicate a significance at the 0.05, 0.01 and 0.001 levels, respectively.

budding yeast[11,57,58] and animal cells[59–61], cell size is likely primarily controlled at the G1/S transition. However, since loss of either G1/S regulation (cycd3;1-3) or G2/M regulation (cdkb1;1/1;2) both resulted in increased cell size, our results indicate that both transitions may be important in regulating cell size in plants. Using our original model, it was not possible to distinguish between mutants affecting G1/S and G2/M since both would be simulated as a reduction in the single pCDK value. We therefore developed the model to incorporate two sequential CDK activities (CDK$_S$ and CDK$_M$) with separate production rates (pCDK$_S$ and pCDK$_M$) to regulate the G1/S and G2/M transitions respectively (Fig. 6a).

We first tested the hypothesis that cell size is regulated at the G1/S transition by making pCDK$_S$, but not pCDK$_M$, dependent on cell size. Thus G1 length is flexible, but S-G2-M takes a constant amount of time to complete (representing sizer followed by timer mechanisms). This model produced size control for a limited range of pCDK$_M$ within which the flexibility in G1 length was sufficient to produce the inverse relationship between cell size and cell cycle length required for homeostasis. Outside this range, cell size was sufficiently large that cells exited G1 immediately after birth. The overall cell cycle length is then determined by the

inflexible, size-independent S-G2-M phase and size control is lost (Fig. 6b,f). A similar situation is found if size control is assumed to occur only at the G2/M transition (timer followed by sizer), although the range of stable pCDK$_S$ values is even narrower due to the shorter length of S-G2-M (Fig. 6c,g). In contrast, if both G1/S and G2/M are sequentially size-dependent (sizer followed by sizer), both phases are flexible in length and cell size control is achieved over a wide range of pCDK$_S$ and pCDK$_M$ values (Fig. 6d,e,h,i). The homeostatic system is therefore more robust if integration of cell size takes place at more than one cell cycle transition and both phases are of flexible length.

**Development of a fluorescent marker for S-G2-M.** To determine experimentally whether both cell cycle phase lengths are indeed flexible, we developed a fluorescent marker for cells in S-G2-M that allows tracking of cell cycle progression in intact tissues. An engineered VENUS fluorescent protein containing the destruction box (DB) sequence from Arabidopsis CYCB1;1 (ref. 62) was placed under the control of the *HISTONE H4* promoter[63], to produce a marker that is switched on at G1/S and degraded during mitosis. Consistent with the expected dynamic

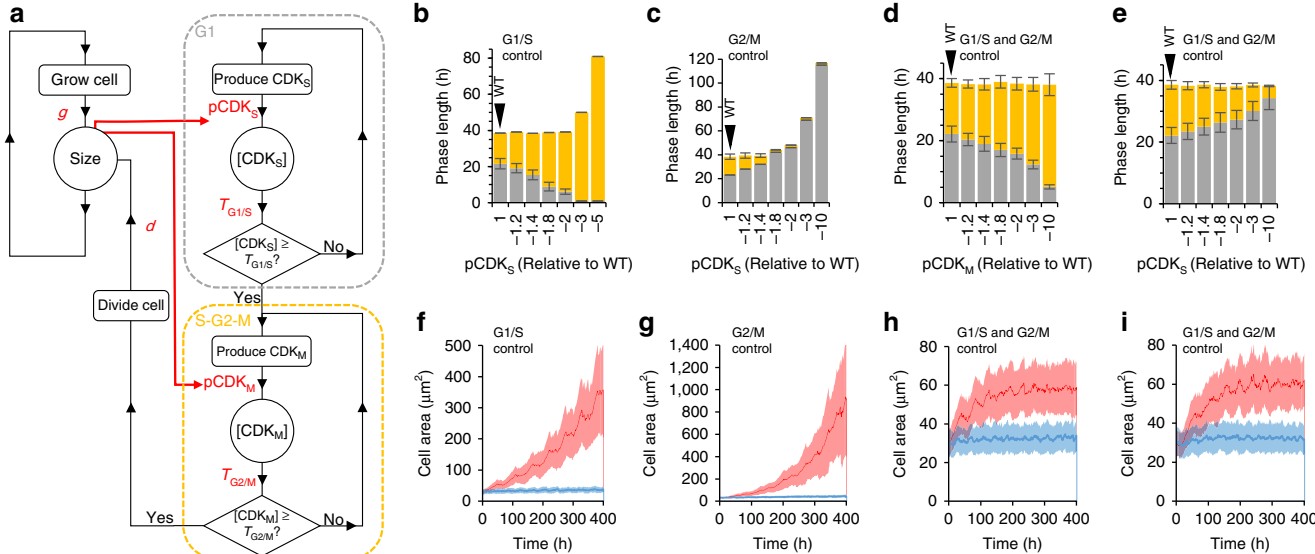

**Figure 6 | Homeostasis of cell size is more robust if cell size is integrated at both the G1/S and G2/M transitions resulting in two phases with flexible lengths.** (**a**) Schematic of the two-transition model of the cell cycle based on two sequential CDK activities. The cell grows with a relative growth rate *g*. The G1/S transition is regulated by the accumulation of CDK$_S$ activity at the rate pCDK$_S$ to a threshold $T_{G1/S}$. The G/M transition is regulated by the accumulation of CDK$_M$ activity at a rate pCDK$_M$ to a threshold $T_{G2/M}$. Once the G2/M transition is passed cell division is triggered and the cell divides with a division ratio *d*. Simulations were initiated with a population of 100 asynchronous cells and observed variation in *g* and *d* were included. (**b**) Model with size controlled at only the G1/S transition. Mean G1 length (grey) and S-G2-M length (yellow) are shown. Error bars show s.d. G1 length is determined by a combination of pCDK$_S$ and cell size, but S-G2-M length is determined solely by pCDK$_M$. Small reductions in pCDK$_M$ (up to 2 fold) do not affect overall cell cycle length since increased S-G2-M length is compensated for by reduced G1 length. Larger reductions in pCDK$_M$ cannot be compensated, since G1 length reaches a minimum and cell cycle length is determined entirely by pCDK$_M$. (**c**) Model with size controlled at G2/M only. Mean G1 (grey) and S-G2-M (yellow) length are shown. Error bars show s.d. S-G2-M length is determined by a combination of pCDK$_M$ and cell size. G1 length is determined solely by pCDK$_S$. Small reductions in cell pCDK$_S$ (up to 1.4 fold) do not affect mean cell cycle length since small increases in G1 length are compensated for by reductions in S-G2-M. Larger reductions cannot be compensated, since S-G2-M length is at a minimum and cell cycle length is determined entirely by pCDK$_S$. (**d,e**) Mean G1 length and S-G2-M length in model simulations in which cell size is controlled at both the G1/S and G2/M transitions. Both G1 and S-G2-M length are determined through a combination pCDK$_S$, pCDK$_M$ and cell size. This model predicts a constant cell cycle length even with large fold reductions in pCDK$_S$ (**d**) or pCDK$_M$ (**e**). Error bars show s.d. X-axes represent the fold reduction in pCDK value used in the simulation relative to pCDK in the wild type simulation. Wild type simulations are shown in the left hand bars. Error bars represent s.d. (**f-i**) Distribution of cell size in model simulations described in parts b-d respectively. Solid lines show mean cell size. Shaded areas represent s.d. (**f**) Model with cell size control at G1/S only. A small (1.2 fold) reduction in pCDK$_M$ (blue) does not affect the ability of the population to maintain a constant mean cell size over time. A larger (3 fold) reduction in pCDK$_M$, (red) leads to loss of cell size control. (**g**) Model with cell size control at G2/M only. A small (1.2 fold) reduction in pCDK$_S$ (blue) does not affect the ability of the population to maintain a constant mean cell size over time. A larger (3 fold) reduction in pCDK$_S$ (red), leads to loss of cell size control. (**h,i**) Model with cell size control at both G1/S and G2/M. Neither small (blue) nor large (red) reductions in pCDK$_M$ (**h**) or pCDK$_S$ (**i**) result in loss of cell size control although cell size is predicted to increase.

expression, VENUS signal was detected for a sustained period leading up to cell division (Supplementary Fig. 6a). To validate whether the appearance of the marker coincided with the G1/S transition, we compared its expression to the pattern of incorporation of EdU, a thymidine analogue commonly used to identify S-phase cells (Fig. 7a–c, Supplementary Fig. 6b–d).

Three populations of cells were identified; presumed G1 cells lacking both EdU and *H4::DB-VENUS*, S-phase cells that have both EdU and *H4::DB-VENUS* and G2-M cells with only *H4::DB-VENUS* (Fig. 7c). Interestingly, predicted G2-M cells were significantly larger than S-phase cells (Fig. 7d), demonstrating that growth does continue during S-G2-M consistent with our proposal that cell size could be relevant to stages of the cell cycle after the G1/S transition. That a smaller increase in size occurs during S-G2-M than in G1 is likely due to the longer relative length of the G1 phase.

**Both G1/S and G2/M transitions are size-dependent.** From time courses using the *H4::DB-VENUS* reporter in developing primordia, we noted that in older primordia both G1 and S-G2-M decreased in length compared to cells in younger primordia (Fig. 7e,f) indicating that neither phase has an absolutely fixed length. An inverse relationship similar to that seen between cell size and cell cycle length was also detected between cell size and both G1 and S-G2-M length (Fig. 7g–j), further supporting the hypothesis that both the G1/S and G2/M transitions are size dependent. The dependency of S-G2-M length on cell size was stronger in younger primordia (p1-p5) (Fig. 7h) than in the larger cells of older primordia (p6-p10) (Fig. 7j), which may reflect that a minimum phase length is eventually reached.

If phase length is dependent on cell size, cells entering a particular phase at a larger size should complete it at an accelerated rate. To test this prediction, we measured phase

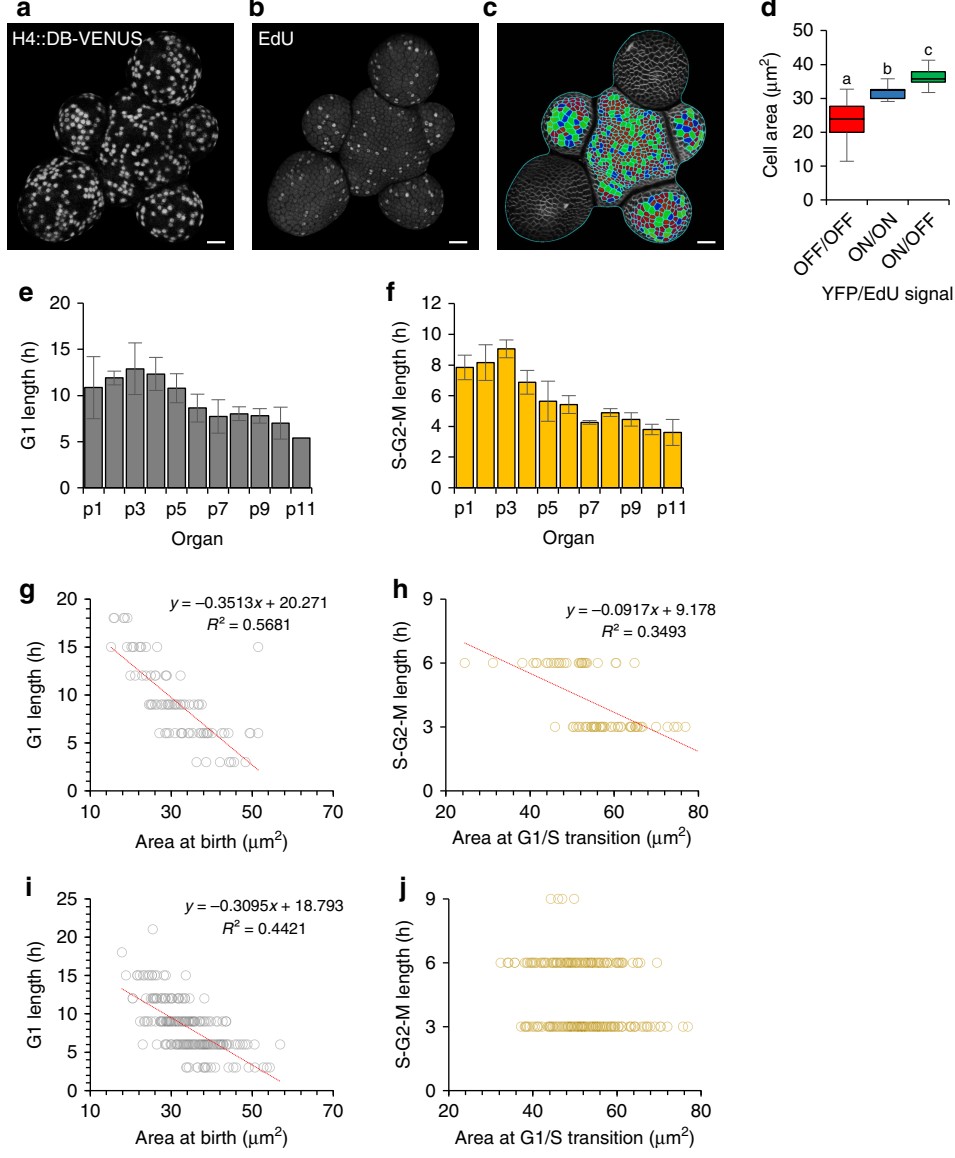

**Figure 7 | G1 length and S-G2-M length are both flexible.** (**a,c**) Comparison of *H4::DB-VENUS* expression and pattern of EdU incorporation. Surface projections of VENUS YFP signal (**a**) and EdU-AlexaFluor488 signal (**b**) from the same stem. Segmented image (**c**) identifying cells with no *H4::DB-VENUS* and no EdU incorporation (red), cells with *H4::DB-VENUS* and EdU (blue) and cells with only *H4::DB-VENUS* (green), predicted to be in G1, S and G2-M respectively. Scale bars represent 20 µm. (**d**) Distribution of cell sizes of predicted G1 (red), S (blue) and G2-M cells (green). Boxes represent the interquartile range, whiskers represent total range. ANOVA, $n = 516$ cells, $F(2) = 118.9$, $P < 0.001$, letters indicate significantly different groups based on Tukey's post hoc analysis. (**e,f**) Mean G1 length (**e**) and S-G2-M length (**f**). Data are grouped by developmental zone. Data shown is the mean from three stems and error bars represent s.d. (**g–j**) Relationship between cell area and phase lengths in young (p1–p5) (**g,h**) and old (p6–p10) (**i,j**) primordia.

length in the *cycd3;1-3* and *cdkb1;1/1;2* mutants, which display increased cell size but whose molecular lesions should affect CDK activity specifically in G1 and G2 respectively. In the *cdkb1;1/1;2* mutant (Fig. 8a,b), S-G2-M phase length was increased compared to wild type, consistent with its delayed G2/M transition (Fig. 8d), and cells were consequentially larger entering G1 (Supplementary Fig. 6e). G1 length was shorter than wild type (Fig. 8c) indicating that the larger cells exited into the subsequent S-phase more rapidly. We conclude therefore that G1 is size dependent.

We next used the *cycd3;1-3* mutant (Fig. 8e,f) affected in the G1-S transition to similarly test if S-G2-M is size dependent. G1 was longer in *cycd3;1-3* cells than in wild type (Fig. 8g) and cells were larger when they entered S-G2-M (Supplementary Fig. 6f). Consistent with S-G2-M phase length also being size dependent, S-G2-M was shorter in larger *cycd3;1-3* cells than in smaller wild type cells (Fig. 8h). Therefore in contrast to previous suggestions that cell size is regulated at G1/S in plants, our experimental results best match the predictions of the model where the progression of both the G1/S and G2/M transitions are size dependent.

## Discussion

Our results uncover the link between the tightly coupled processes of growth and division in an actively dividing plant tissue. We propose that cell size in the SAM is regulated by an intrinsic balance between cell growth and cell division, which determines the size at which cells divide. Despite the presence of extracellular signaling molecules and mechanical constraints that

are likely to restrict cell size within tissues, we find that cell size control in the Arabidopsis SAM is very similar to that described in yeast, a single-celled, free-living organism in which cell size can primarily be viewed as an adaptation to the environment. Furthermore, changes in cell size observed during organogenesis may be explained as an emergent property of increasing RGR within the system. Therefore, rather than requiring specific developmental mechanisms to alter cell size during development, plants may be utilizing the ancestral relationship between growth and division not only to achieve cell size homeostasis within dividing tissues, but also to re-set cell size during organogenesis.

Central to the mechanism of control is the cell-size dependent progression of the cell cycle, mediated either by size-dependent accumulation of CDK activity or size-dependent CDK thresholds. Although such models have long been proposed, identifying the molecular mechanisms underpinning them has proven more difficult. In budding yeast the accumulation of the G1 cyclin Cln3, the closest yeast homologue to plant CYCDs, and the dilution of transcriptional inhibitor Whi5, analogous in function to plant RETINOBLASTOMA-RELATED (RBR)[64] a downstream target of CDKA-CYCD complexes, have been shown to be critical for size-dependent cell cycle progression in yeast. However, differential degradation of cell cycle regulators and interdependency between G1/S and G2/M components are also likely to be important. CYCD and CDKB1 proteins together with their interactors and downstream effectors will therefore need to be quantified in detail throughout the cell cycle in order identify the exact molecular mechanisms acting in plants. The volume of cell compartments such as the nucleus, and the localization of

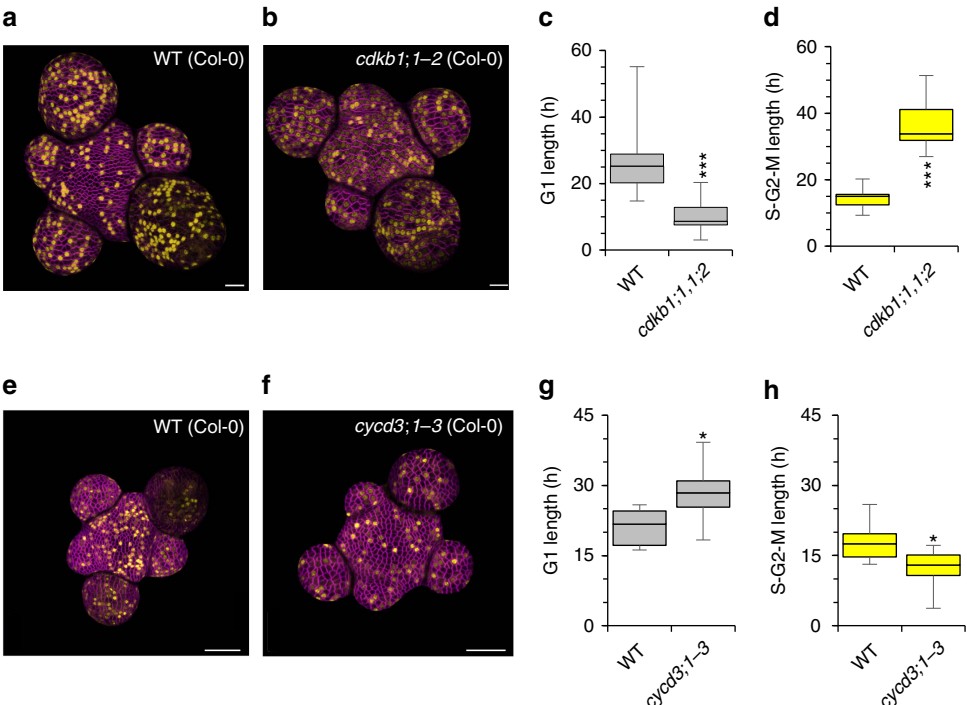

**Figure 8 | Mutant analysis suggests length of G1 and S-G2-M is size dependent.** (**a,b**) Surface projections of WT (Col-0) (**a**) and *cdkb1;1/1;2* double mutant plants (**b**) expressing *H4::DB-VENUS*. VENUS signal is shown in yellow. Cell membranes are shown in magenta. Scale bars represent 20 μm. (**c**) Distribution of mean G1 length in WT (Col-0) and *cdkb1;1/1;2* plants. Data represent mean values from 13 and 11 plants respectively. (*t*-test, *t* = 4.6858, df = 19.854, *P* = 0.0001) (**d**) Distribution of mean S-G2-M length in WT (Col-0) and *cdkb1;1/1;2* plants. Data represent mean values from 13 and 11 plants respectively. (*t*-test, *t* = −4.6858, df = 19.854, *P* = 0.0001) (**e,f**) Surface projections of WT (Col-0) (**e**) and *cycd3;1-3* triple mutant plants (**f**) expressing *H4::DB-VENUS*. VENUS signal is shown in yellow. Cell membranes are shown in magenta. Scale bars represent 50 μm. (**g**) Distribution of mean G1 length in WT (Col-0) and *cycd3;1-3* plants. Data represent mean values from 9 and 7 plants respectively. (*t*-test, *t* = 2.6144, df = 9.129, *P* = 0.02775) (**h**) Distribution of mean S-G2-M length in WT (Col-0) and *cycd3;1-3* plants. Data represent mean values from 9 and 7 plants respectively. (*t*-test, *t* = −2.6144, df = 9.129, *P* = 0.02775) *, **, *** indicate a significance at the 0.05, 0.01 and 0.001 levels respectively. Error bars show total range.

molecules within them, for example to binding sites on DNA, will also need to be investigated to distinguish between mechanisms; an activator accumulator mechanism requires a constant nuclear volume or number of binding sites[65] to titrate CDK activity against, whereas an inhibitor-dilution mechanism requires an increase in volume[19].

In contrast to many systems where G1 has been found to be flexible in length and S-G2-M fixed, we found evidence of size-dependent control at both the G1/S and G2/M transitions consistent with work in fission yeast[66]. However, in comparison to wild type fission yeast grown under nutrient rich conditions, which has a very short G1 and effectively depends upon the G2/M transition for cell size control[8,12,66], phase lengths in wild type Arabidopsis SAM cells were more balanced and more closely resemble fission yeast grown under nutrient poor conditions. Maintaining two flexible phases may create greater control of cell cycle length and be important in allowing cell size to be coordinately regulated in growing tissues in which the cell cycle is not synchronized. Intriguingly, our results indicate that it is the plant specific, mitotic CDKB that pushes cells from a fission-yeast like cycle with short G1 into a cycle with a longer G1.

Finally we note that since the system described here is dynamic, cell size at division, cell cycle length and the lengths of the G1 and S-G2-M phases of the cell cycle are all flexible and may be responsive to a range of intrinsic and extrinsic signals. Cell size at division should therefore be regarded not as a fixed parameter that functions to produce arbitrary subdivisions of tissue space, but rather as an emergent property that allows for flexibility in cell size and creates an additional level of response through which environmental and developmental signals are integrated into plant tissue growth and structure.

## Methods

**Plant material and growth conditions.** pPIN1::PIN1-GFP (Col-0)[67], cycd3;1-3 (Col-0)[54], cdkb1;1/1;2 (Col-0)[55], p35s::CYCD3;1 (Ler)[45] have been described previously. Seeds were stratified in the dark at 4 °C for 48 h then germinated on solid growth medium and grown for seven days (continuous light at 60 μmol m$^{-2}$ s$^{-1}$, 21 °C) before transferring to soil and growing to the floral transition (16 h light at 150 μmol m$^{-2}$ s$^{-1}$, 8 h dark, 21 °C). For low light growth conditions, light intensity was reduced to 10 μmol m$^{-2}$ s$^{-1}$. SAMs were dissected once the inflorescence stem was > 0.5 cm in height. Dissection and culture of stems was carried out as described previously[68]. Unless otherwise stated SAM culture medium contained 1% sucrose. Where sucrose concentration was reduced, sucrose was replaced with an equal molarity of mannitol. For time course experiments stems were allowed to recover overnight after dissection before beginning imaging. For light intensity experiments, stems were imaged immediately after dissection.

**Confocal microscopy.** For analysis of wild type stems over 96-hour and 30-hour time courses, expression of pPIN1::PIN1-GFP was used to identify cell membranes. For mutant analysis, stems were immersed in 33 μg μl$^{-1}$ FM 4–64 (Molecular Probes) for 30–60 s then incubated for a further 5 min before imaging. Confocal stacks were taken using a Zeiss 780 Meta confocal microscope with × 40 water dipping objective. For detailed wild-type time courses, stems were imaged every 3 or 8 h for a total of 96 or 30 h, respectively. For mutant analysis two images were taken 24 h apart. Between imaging, stems were returned to the growth chamber.

**Image processing and analysis.** Confocal stacks were analysed using Morpho-GraphX (ref. 40). Signal from layer one (L1) of the SAM was projected onto a 2.5D surface representing the outer surface of the 3D meristem. Signal from cell membranes was used to segment the projected surface into individual cells. Since L1 thickness was consistent (Supplementary Fig. 1a,b), outer surface area was found to be a good proxy of cell volume (Supplementary Fig. 1c). Lineage information was manually annotated through pairwise comparisons of consecutive time points using lineage tracking tools, then compiled along with outer cell surface area in an SQLiteMan database. Developmental stages were identified according to morphological landmarks and timecourse information; the central zone was estimated as the area between outgrowing primordia, p1 was identified as the first primordia in which PIN1 veination was identified, primordia 2–10 were numbered according to the assumed order of phyllotaxy and size of organ, incipient primordia i1-i3 were predicted relative to the position of the outgrowing primordia. Organ stage was recorded at birth for measurements of cell cycle length and G1 length and at the G1/S transition for measurements of S-G2-M length. RGR was

calculated using the following formula $RGR = (\ln A_1 - \ln A_0)/t_1 - t_0$ where $A_0$ is the outer cell surface area at the beginning of the experiment ($t_0$) and $A_1$ is the outer cell surface area after 24 h ($t_1$). To calculate cell cycle length and phase length, the equation was rearranged to calculate the amount of time required to increase in size from mean birth size to the mean division size (for cycle length), mean birth size to mean G1/S transition size (for G1 length) and from mean G1/S transition size to mean division size (for S-G2-M length) based on the mean measured RGR for that stem. Cells that had undergone the G1/S transition were identified through the expression of H4::DB-VENUS in a cell, where expression had not previously been detected. Statistical analyses were performed in R. For cell size comparisons many cells were measured from each stem, therefore distributions were analysed using a generalized linear model, including stem as a random factor. To evaluate whether the data met the assumptions of the model, diagnostic plots were used to confirm that the residuals exhibited homogeneity, normality and independence. Cell cycle length and phase length were calculated as a stem average, and compared by T-test for mutant versus wild type analysis and by ANOVA for organ stage analysis. Sample sizes were determined based on the effect sizes observed in pilot experiments. Any stems that were damaged during the time course were excluded from analysis.

**Generation of *pH4::DB-VENUS* reporter.** The DNA construct was built using a three-way Gateway (Invitrogen) reaction according to the manufacturer's instructions. The promoter fragment of the HistoneH4 gene, as described in ref. 63 was cloned into pDONR L4-R1, a 357 bp fragment encoding the N terminal sequence of CYCLIN B1;1 (ref. 62) was cloned into pENTR SD TOPO and VENUS was cloned into pDONR R2-L3. Binary vector pB7m34GW[69], containing R4 and R3 recombination sites and BAR gene for plant selection, was used as the destination vector and for plant transformation. The construct was introduced into wild type (Col-0) plants by floral dipping.

**Detection of DNA synthesis.** Dissected SAMs containing *pH4::DB-VENUS* were imaged as described above. Stems were transferred to solid media containing 20 μM 5-ethynyl-2′-deoxyuridine (EdU) (Molecular Probes) and submerged in sterile water containing 20 μM EdU for 5 min. Submersion liquid was removed and stems incubated for a further 3 h before transferring to fresh, unsupplemented media and re-imaging. Immediately after imaging, stems were fixed by incubating in a solution of 3.6% formaldehyde, 0.01% Triton X100 in PBS (pH 7) for at least 1 h. Stems were rinsed three times in water before performing the Click-iT (ThermoFisher Scientific) detection reaction. For efficient detection[56] stems were incubated for one hour in detection solution one (10 μM Alexa 488 Azide, 100 mM Tris pH 8.5) followed by 30 min in detection solution two (10 μM Alexa 488 Azide, 100 mM Tris pH 8.5, 1 mM CuSO$_4$, 100 mM Ascorbic Acid).

**One transition model implementation.** Cell division models were run iteratively in MATLAB. Each time step ($t$), equating to 1 h, cell size is recalculated according to the growth rate, $g$ (1), then CDK activity is produced according to the production rate pCDK (2).

$$\text{Size}_{t(n)} = \text{Size}_{t(n-1)} + \left(\text{Size}_{t(n-1)} \times g \times (t_{(n)} - t_{(n-1)})\right) \quad (1)$$

$$\text{CDK}_{t(n)} = \text{CDK}_{t(n-1)} + \left(\text{pCDK} \times (t_{(n)} - t_{(n-1)})\right) \quad (2)$$

Total CDK activity is compared to the CDK threshold value $T_{\text{Division}}$. If the total CDK activity is greater than $T_{\text{Division}}$, mitosis is triggered. Mitosis takes a single time step to complete. The cell divides according to a division ratio $d$ to give two daughters of appropriate sizes. At birth each cell is assigned a value for $d$, either fixed at 50 for models without variation or drawn from a normal distribution based on our measurements for models with variation (Supplementary Fig. 2b). When the cell divides, this preassigned value is used to determine the area of the first daughter as a percentage of the parent area. The area of the second daughter is then calculated by subtraction.

For size-dependent models either pCDK is made proportional to cell size (3) or $T_{\text{Division}}$ is made inversely proportional to cell size (4).

$$\text{CDK}_{t(n)} = \text{CDK}_{t(n-1)} + \left(\text{Size}_{t(n)} \times \text{pCDK} \times (t_{(n)} - t_{(n-1)})\right) \quad (3)$$

$$T_{\text{DivisionS}} = T_{\text{Division}} \times 1/\text{Size} \quad (4)$$

The parameter values used in the simulations for $g$, pCDK, $T_{\text{Division}}$ and $d$ are given in Supplementary Table 1. Values for $g$ and $d$ are based on our experimental observations. $g$ is constant and independent of cell size, consistent with our observations that size accounts for very little of the observed variation in RGR within a developmental zone (Supplementary Fig. 1d). Introducing a size-dependent RGR term based on our regression slope did not affect our conclusion that an inverse relationship between cell size and cell cycle length is required for size control, or that CDK activity is central to this relationship. It does however produce a small increase in cycle length in large celled mutants (cycd3;1–3 and cdkb1;1/1;2) and a slight decrease in cell cycle length in small celled simulations (35s::CYCD3;1) (Supplementary Fig. 7). The ratio between pCDK and $T_{\text{Division}}$, which determines cell cycle length, was set for each model variation as follows; first the starting value for $T_{\text{Division}}$ was set at an arbitrary value, then the value of pCDK

required to produce an increase in cell area from 20 to 40 μm$^2$ per cell cycle given the value of $T_{Division}$ was determined (Supplementary Table 2). The effect of altering this 'wild type' ratio was then investigated by altering pCDK, while keeping $T_{Division}$ constant.

For model simulations including observed variation in RGR and division ratio, cells were assigned a value of $g$ and $d$ selected at random from a normal distribution of values based on the measured mean and s.d. Simulations were initiated with 100 asynchronous cells. The population was restricted to 100 cells by removing randomly selected cells when the population exceeded 100.

**Two transition model implementation.** The two transition model is an extension of the one-transition model, where instead of triggering cell division, attainment of the first threshold ($T_{G1/S}$) triggers the initiation of a second CDK regulated phase. When the second threshold ($T_{G2/M}$) is met cell division is triggered. Both phases are simulated as described above. Parameter values for $g$ and $d$ were as used in the one-transition model. The initial 'wild type' ratios between pCDK$_S$ and pCDK$_M$ and their respective thresholds were determined as described above. To capture the meaningful relationship between $T_{G1/S}$ and $T_{G2/M}$, initial threshold values were set based on relative measurements of CDK activity at the G1/S and G2/M transitions in synchronized Arabidopsis cell cultures[70]. Wild type ratios produced an average increase in cell area from 20 to 40 μm$^2$ per cell cycle and G1 length representing 60% of the total cell cycle length These 'wild type' ratios, were altered by changing pCDK$_S$ or pCDK$_M$, but keeping $T_{G1/S}$ and $T_{G2/M}$ constant.

**Code availability.** Matlab scripts model simulations are available at https://github.com/Angharad-Jones/CellCycleModel. Scripts used in data analysis are available from the authors upon request.

**Data availability.** All relevant data are available from the authors on request.

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

## Acknowledgements

We would like to thank Teva Vernoux, Christophe Godin and Meirion Jones for advice and discussions, Pradeep Das, Sophy Chamot and Géraldine Brunoud for training in meristem dissection and imaging and Angela Marchbank and Joanne Kilby for expert technical support. The project was supported within the ERA-NET in Systems Biology (ERA-SysBio + ; project iSAM) funded with support of the European Commission, by the BBSRC (grants BB/I004661/1; BB/J009199/1) and by the Cardiff University Synthetic Biology Initiative.

## Author contributions

A.R.J. performed experiments and simulations, analysed data and wrote the manuscript. M.F.-V. produced image analysis software used for essential preliminary analyses. S.P.W. designed database structure and produced software for storage and analysis of time course data. R.S.S. provided access, training and supervision in the use of Morpho-GraphX software. J.T., W.D. and J.A.H.M. designed the study, discussed results and interpretation and contributed to manuscript preparation.

## Additional information

**Competing interests:** The authors declare no competing financial interests.

