## [Peer Review File · Nature Communications]

Reviewers' Comments:

Reviewer #1 (Remarks to the Author)

The authors have addressed my concerns and this paper should be published in Nature Communications.

Reviewer #2 (Remarks to the Author)

This manuscript has greatly improved and now presents an excellent data set and mathematical model in a clear way, showing that plant cells exert cell-autonomous control of cell size by making both G1/S- and G2/M-CDK activity proportional to cell size. I now enthusiastically support publication of this paper after a few minor quibbles have been dealt with:

1. line 243-246: this is a cryptic sentence, please clarify.
2. I appreciate the explanation of units used in the Matlab code, but still have an issue about equation 4: If Size is measured in area units, Tdivision is no longer defined in arbitrary units but in area⁻¹, right?

Reviewer #3 (Remarks to the Author)

This manuscript contains a series of very elegant and well-designed experiments to answer the question of how cells coordinate cell size and cell cycle progression to control cell size at the moment of cell division. The added value of this work is that the authors use a strategy to obtain measurements directly in a growing organ of a multicellular organism. The topic is of high relevance for a wide audience and is addressed with a combination of high-resolution time-lapse imaging and computer modeling. The manuscript is a revised version where the authors have addressed a number of concerns on a previous version. In most cases their reply and/or the new data provided are satisfactory. The text reads smoothly and the general conclusions (the highly dynamic and flexible nature of the coordination between cell size and cell division) are supported by the results provided.

Specific comments.

1. Line 249. Not sure that all the information mentioned here is actually provided in the Suppl Fig 4d-f (Tdivision?).
2. Suppl Fig 6. If I understood this correctly, I consider that the timing used in these experiments (30min, 8h) does not provide sufficient resolution as to measure/define the G1/S transition, which is a rather fast event.
3. Fig. 6b-e. I suggest rewrite these experiments to explain how increases in CDKm or CDKs lead to increases in G2 or G1 lengths, respectively.
4. I perceive a significant discrepancy in the measurements of cell cycle length presented in Suppl. Fig. 6 and Fig. 7. Based on images in the SF6 the time between G1/S and G2/M is ~ 2.25 h (images taken every 15min). This seems quite different from measurements in other parts of study, e.g. Fig 7f for the s-G2/M. Additionally, adding the lengths presented in Fig 73 and 7f does not coincide, even approximately, with the total cell cycle length (of the same cell types, p6, p7 for example) presented in Fig 2f (red lines). This needs a clarification to explain the differences in cell cycle length in the two different sets of experiments.

Reviewer #4 (Remarks to the Author)

I have read carefully this new version of the manuscript and have assessed the responses to my previous review. I think that the revised manuscript has improved and all my concerns have been satisfactorily addressed.

I would suggest that this manuscript describing that cell size and cell division are coordinated in the Arabidopsis shoot apical meristem is now ready for publication in Nature Communications.

Reviewer #1 (Remarks to the Author):

The authors have addressed my concerns and this paper should be published in Nature Communications.

Reviewer #2 (Remarks to the Author):

This manuscript has greatly improved and now presents an excellent data set and mathematical model in a clear way, showing that plant cells exert cell-autonomous control of cell size by making both G1/S- and G2/M-CDK activity proportional to cell size. I now enthusiastically support publication of this paper after a few minor quibbles have been dealt with:

1. line 243-246: this is a cryptic sentence, please clarify.

This concept has been clarified in the main text (lines 244-249) and in the legend of supplementary figure 3.

2. I appreciate the explanation of units used in the Matlab code, but still have an issue about equation 4: If Size is measured in area units, T_{division} is no longer defined in arbitrary units but in area-1, right?

Yes, in the case that T_{Division} is dependent on Size ($T_{\text{Division-S}}$), it will be defined by Arbitrary Units per μm^2 . To clarify this, we have included an additional entry in supplementary table 1.

Reviewer #3 (Remarks to the Author):

This manuscript contains a series of very elegant and well-designed experiments to answer the question of how cells coordinate cell size and cell cycle progression to control cell size at the moment of cell division. The added value of this work is that the authors use a strategy to obtain measurements directly in a growing organ of a multicellular organism. The topic is of high relevance for a wide audience and is addressed with a combination of high-resolution time-lapse imaging and computer modeling. The manuscript is a revised version where the authors have addressed a number of concerns on a previous version. In most cases their reply and/or the new data provided are satisfactory. The text reads smoothly and the general conclusions (the highly dynamic and flexible nature of the coordination between cell size and cell division) are supported by the results provided.

Specific comments.

1. Line 249. Not sure that all the information mentioned here is actually provided in the Suppl Fig 4d-f (T_{division} ?).

The negative result referred to in the text has now been added (Supplementary Fig. 4g-i).

2. Suppl Fig 6. If I understood this correctly, I consider that the timing used in these experiments (30min, 8h) does not provide sufficient resolution as to measure/define the G1/S transition, which is a rather fast event.

With regard to the selection of time intervals for the experiments shown in Supplementary Figure 6, we have faced a trade-off between the total length of the time course, the number of frames that can be taken without damaging the specimen and the number of specimens that need to be imaged for proper analysis. We believe that our choices are appropriate for the purposes of the different experiments. The time course shown in Supplementary Figure 6a (30 min intervals, 7 hours total) is included to demonstrate the expression of the new reporter that we have developed in cells leading up to division. It is not intended to show an entire cycle and therefore we do not draw any conclusion from this timecourse regarding G1 length or the timing of G1/S. We have clarified this in the text (lines 332-335).

3. Fig. 6b-e. I suggest rewrite these experiments to explain how increases in CDK_m or CDKs lead to increases in G2 or G1 lengths, respectively.

As suggested we have rewritten the explanations of these experiments (lines 964-1007).

4. I perceive a significant discrepancy in the measurements of cell cycle length presented in Suppl. Fig. 6 and Fig. 7. Based on images in the SF6 the time between G1/S and G2/M is ~2.25h (images taken every 15min). This seems quite different from measurements in other parts of study, e.g. Fig 7f for the s-G2/M.

We apologise for an error in the legend. Images in this time course were in fact taken at 30 minute intervals. The full time course shown is therefore seven hours in total and the time between G1/S (T2) and the completion of division (T11) for the indicated cell is six hours, consistent with S-G2-M lengths as presented in Figure 7f.

The information in the figure legend has been corrected.

Additionally, adding the lengths presented in Fig 7g and 7f does not coincide, even approximately, with the total cell cycle length (of the same cell types, p6, p7 for example) presented in Fig 2f (red lines). This needs a clarification to explain the differences in cell cycle length in the two different sets of experiments.

Comparison of the different experiments as described by the reviewer is problematic for a number of reasons. First, the graph in Figure 2f shows the average phase length using data from three stems. In contrast the graphs shown in Figure 6e and f show the mean phase length of cells from a single stem. To allow for better comparison, we have replaced the graphs in Figure 7 with graphs also showing the mean phase lengths calculated from three stems (Figure A).

Second, it is difficult to justify how one should sum the values in Figure 7 to produce total cell cycle length comparable to those in Figure 2. In the experiment shown in Figure 2, cells were assigned to an organ at birth, but the cycle was not necessarily completed while the organ was still at the same stage of development. Indeed it is likely that many cells that were born in an organ classified as p6 actually underwent G1/S transition when the organ would have been classified as p7. Therefore rather than summing G1 and S-G2-M from p6 as the reviewer suggests ($8.64 \text{ hr} (\pm 1.52) + 5.43 \text{ hr} (\pm 0.58) = 14.07 \text{ hr} (\pm 1.63)$), a more realistic estimate is likely to be given by summing G1 from p6 and S-G2-M from p7 ($8.64 \text{ hr} (\pm 1.52) + 4.26 \text{ hr} (\pm 0.13) = \mathbf{12.9 \text{ hr} (\pm 1.53)}$). Similarly to approximate cycle length in p7, G1 from p7 and S-G2-M from p8 should be summed ($7.73 \text{ hr} (\pm 1.80) + 4.90 \text{ hr} (\pm 0.25) =$

12.63 hr (± 1.82)). This evaluation brings these estimates much closer to the values shown in Figure 2 (p6 **10.98 hr (± 0.81)**, p7 **11.69 hr (± 1.08)**) and given the different approaches used in the two figures we believe they are consistent.

We note also that the major conclusion drawn from Figure 2f, that cell cycle length decreases as developmental stage increases, is consistent between experiments irrespective of the assumptions made when summing the phase lengths.

We have added a description of how cells were assigned organ stage values in the methods section (lines 482-484).

Reviewer #4 (Remarks to the Author):

I have read carefully this new version of the manuscript and have assessed the responses to my previous review. I think that the revised manuscript has improved and all my concerns have been satisfactorily addressed.

I would suggest that this manuscript describing that cell size and cell division are coordinated in the Arabidopsis shoot apical meristem is now ready for publication in Nature Communications.

Reviewers' Comments:

Reviewer #3 (Remarks to the Author)

Authors have addressed satisfactorily all my concerns.